

**Estimations of soil fertility in physically degraded soils through selective accounting of fine**
**earth**
Mavinakoppa S. Nagaraja[1], Ajay Kumar Bhardwaj [2]*, G.V. Prabhakara Reddy[3], Chilakunda A.
Srinivasamurthy[3], Sandeep Kumar [4]
[1] College of Horticulture, University of Horticultural Sciences, Bagalkot-587102, India
[2] Division of Soil and Crop Management, ICAR-Central Soil Salinity Research Institute, Karnal-
132001, India
[3] Department of Soil Science, University of Agricultural Sciences, Bangalore -587165, India
[4] Department of Plant Science, South Dakota State University, Brookings, SD 57007, USA
*_Correspondence to_: Ajay Kumar Bhardwaj (ak.bhardwaj@icar.gov.in)




**Abstract**
Soil fertility and organic carbon (C) stock estimations are crucial to soil management especially
that of degraded soils, for productive agricultural use and in soil C sequestration studies.
Currently, estimations based on generalized soil mass (hectare-furrow basis) or bulk density
(BD) basis are used which may be suitable for normal agricultural soils but not for degraded
soils. We measured soil organic C, available nitrogen (N), available phosphorus (P) and available
potassium (K), and estimated stocks using three methods: (i) generalized soil mass (GSM, 2
million kg ha$^{-1}$ furrow soil), ii) bulk density based soil mass (BDSM) and (iii) the proportion of
fine earth volume (FEV) method, for soils sampled from physically degraded lands in Eastern
Dry Zone of Karnataka State in India. Comparative analyses using these methods revealed that
the soil organic C, and N, P and K stocks determined by using BDSM were higher than those by
GSM method. The soil organic C values were the lowest in the FEV method compared to the
other two methods. The GSM method overestimated soil organic C, N, P and K by 9.3-72.1%,
9.5-72.3%, 7.1-66.6% and 9.2-72.3 %, respectively, compared to FEV based estimations for
physically degraded soils. The differences among the three methods of determinations were
lower in soils with low gravel content and increased with increase in gravel volume. There was
overestimation of soil organic C and soil fertility with GSM and BDSM methods. A
reassessment of methods of estimation was, therefore, attempted to provide fair estimates for
land development projects in degraded lands.

**1 Introduction**
Precise soil-fertility and crop-nutrition assessments are important for sustainable productivity in
agricultural lands, especially in soils with inherent low carbon or high degradation. Mass-volume



relationships are crucial in estimating soil fertility (Hartemink, 2006) and for developing
reclamation plans. In recent years, these relationships have been used in soil carbon (C) stock
estimations to assess the sink-source potential of soils for atmospheric carbon dioxide (Lorenz
and Lal, 2005) and responses from management under different climatic conditions (Zubrzycki
*et al.*, 2014; Srinivasarao *et al.*, 2014; Lozano-Garcia and Parras-Alcantara, 2014; Parras-
Alcantara *et al.*, 2015, Kaleem Abbasi *et al.*, 2015). The calculations are based on the soil
organic C and nutrient concentrations assessed for a few grams of soil are translated later to a
'generalized soil mass' (GSM) of 2 million kg ha$^{-1}$ to a depth of 15 cm with an often assumed
soil bulk density (BD) of 1330 kg m$^{-3}$. This assumed GSM refers mostly to soils that generally
have equal proportions of solids and void space, with negligible amount of gravel. In these
estimations, therefore, importance is given to total mass instead of  the actual soil mass (based on
field BD) or the proportional volume of fine soil (without gravel portion). However, in
physically degraded soils, gravel content is at least 15% of the total soil volume (Soil Survey
Division Staff, 1993). Hence, void space occupied by gravel can hardly be ignored.  High gravel
content can affect the accuracy of soil fertility estimations in degraded soils if estimations are
based only on GSM.   However, given the importance of reclaiming degraded soils and
exploiting them for agriculture or any other land use under climate change mitigation projects
(Mishra *et al.,* 2015), accurate estimation of soil fertility becomes important, both for location-
specific nutrient applications and assessment of  $CO_2$ sink-source potential (Hartemink, 2006).
Precise quantitative assessments help land developers and farmers to select management plans
best suited to available soil resources, as well as to get realistic responses from management
(Karlen *et al*., 2003, Parras-Alcantara *et al.*, 2015).



Estimations with GSM may not be realistic for all soils as BD values are not the same
(Arvidsson, 1999; Hartemink, 2006). Alternatively, the use of undisturbed field BD values in
nutrient estimations appears more pragmatic. But increase in gravel content, as seen in degraded
soils, adds to the field BD values which can further overestimate soil fertility (Nagaraja and
Srinivasamurthy, 2009). In reality, degraded soils have greater proportion of coarse fragments as
the fine fractions are physically eroded. This increase in proportion of coarse fragments in soil
reduces the volume of space effectively available for water and nutrient retentions, and also for
plant root explorations (Nagaraja and Srinivasamurthy, 2009; Rao and Jessy, 2007; Grewal et al.,
1984). In other words, the quantity of soil organic C and potentially available nutrients for plant
uptake get reduced with increase in volume of coarse fragments. Therefore, soil organic C and
nutrients are generally expected to decline with increase in gravel content. This suggests that
their estimations would be more realistic if it is based on the fine earth volume instead of a
generalized soil mass or BD based estimations.
Eastern dry zones of Karnataka state in South India are considered as bioresource-
deficient zones (Ramachandran *et al*., 2004). Almost 50% of rain is received in Kharif season
(July-October). The soils are coarse textured with predominance of gravel. Management of soil
fertility in the soils of the region is crucial to support good productivity under water stress which
is prevalent during most parts of the year. Soil fertility estimations are crucial, therefore, to plan
fertilizer inputs. Hypothetical estimations suggested that the GSM method would overestimate
nutrient content for the degraded soils of this region and, therefore, the current practice of using
this method needs to be modified. However, this needs validation with actual field data before
deciding the methodology for nutrient estimations. Therefore, this study was undertaken, using
field sampling of degraded soils from diverse landscapes in Eastern Dry Zone of Karnataka State



in India, to evaluate the effect of GSM, BD and fine earth volume-based estimation methods for
the assessment of soil C and nutrient stocks for these physically degraded soils. These
estimations are crucial to land-use and land development programs most often implemented in
resource deficient zones, like the one under reference, in other parts of the world.
**2 Methods**
**2.1    Study area**
The study area consisted of 18 sites in Eastern Dry Zone of Karnataka state in India, covering
parts of Bangalore, Kolar and Tumkur districts (Fig. 1). The annual rainfall in the area ranges
from 679 to 889 mm. The predominant soils of the region are red soils overlying granite from
which they are formed, with texture from gravely sandy loam to sandy clay loam (Soils of
Karnataka, 1998). A preliminary survey was carried out initially in the entire Eastern Dry Zone.
Available information was gathered from various secondary sources such as Departments of
Statistics, Agriculture, and Forests to locate the existing physically degraded (eroded) lands in
this red soil region. Based on the existing secondary information, a physical survey was carried
out later by traversing through the region to choose 18 different sites for soil sampling. The
locations of the sampling sites are depicted in Figure 1. The exact sampling locations were fixed
after giving regard to the visible features such as vegetation, magnitude of erosion and surface
gravel content. Samples of agricultural and non-agricultural soils at the same sites were collected
to include a wide range of gravel proportions.

**2.2    Collection of soil samples for comparative analysis**





Surface soil samples up to 15 cm depth were collected from lands exposed to different
magnitudes of erosion. The samples were carried to laboratory to analyze the volumetric
distribution of fine earth and coarse fragments (gravel) in the soil. The samples were air dried
and separated into coarse fragments ($> 2$ mm) and fine earth ($< 2$mm) by sieving. These
separates were weighed and the proportion of coarse fragments was derived on weight basis. The
coarse fragments in the soils were of granite-gneiss origin. Coarse fragments retained on the
sieve were washed with a jet of water and their respective volumes were determined by
volumetric water displacement method (Jalota *et al.,* 1998). Finally, the volume of coarse
fragments was deducted from the bulk soil volume to assess the proportional volume of fine
earth.

11           The undisturbed core method was used for determination of bulk density (Jalota *et al.,*

1998). The SOC was determined using the wet combustion method (Mebius, 1960; Schumacher,
2002), and was then used in estimating SOC stocks for different mass-volume relationships.
Available nitrogen (N) was determined according to Subbiah and Asija, 1956, phosphorus (P)
was determined colorimetrically using a spectrophotometer (Olsen *et al.,* 1954), and potassium
(K) was determined following the method used by Hanway and Heidel (1952).
**2.3    Statistical analysis**
The statistical analysis of all parameters was done using SAS (2009; SAS Inc., Cary, NC, USA).
All parameters were tested using a one-way analysis of variance (ANOVA) and separation of
means was subjected to Tukey's honestly significant difference test (Steel and Torrie, 1960).
Correlation analysis was conducted to identify relationships between the measured parameters.
All tests were performed at 0.05 significance level.



## 3 Results and Discussion

### 3.1 Effect on soil organic C stock

A hypothetical depiction (Fig. 2a) shows the influence of gravel on soil organic C in degraded lands when 3 different soil mass-volume relationships namely, 2.0 million kg furrow soil (GSM; Scenario 1), BD based soil mass (BDSM, Scenario 2), and the proportion of fine earth volume (FEV, Scenario 3) were used. The soil organic C did not vary with gravel volume when a GSM of 2 million kg was used in the estimations. However, soil organic C estimations based on BD increased with gravel content. Contrastingly, estimations based on the fine earth volume showed a decline in soil organic C. The soil fertility values were also expected to exhibit similar trend as estimation methodology remains the same. Hypothetically, the soil organic C and fertility estimated values could be of the order Scenario 2 > Scenario 1 > Scenario 3.

Analyses based on field collected samples revealed a decline in soil organic C with increase in gravel per cent in all the three methods (Fig. 2b). The soil organic C stocks based on BD (Scenario 2) were found higher than the present GSM method of estimations (Scenario 1). However, the fine earth portion based soil organic C stocks (Scenario 3) remained lower than the other two estimations. The inverse relationships between the soil organic C stocks and the gravel content in field samples may be attributed to the loss of silt and clay during erosion (Lal, 1995; Rezai and Gilkes, 2005). The accumulation of gravel in the soil layer indirectly reflected the extent of loss of fine soil (Grewal et al., 1994; Lal, 1995). The magnitude of differences among three estimates was found to be the least in soils with low gravel content, and it increased with increase in gravel volume.



In case of hypothetical estimates, the present method of soil organic C stock estimations
(GSM, Scenario 1) remained the same at different gravel volumes. Contrastingly, with field
samples it declined with increase in gravel content. This is due to the fact that the GSM of 2
million kg and soil organic C of 0.5% were used in hypothetical estimations, whereas the soil
organic C content declined with gravel volume in field conditions. In case of BD based
estimations (BDSM, Scenario 2), the hypothetical soil organic C stock values increased with
gravel content, but in contrast it decreased in the field soil samples. The increased soil mass due
to increase in BD values with fixed soil organic C content (0.5%) enhanced the hypothetical soil
organic C stocks, while the decrease in soil organic C content in the field samples resulted in
their reduction. The soil organic C estimations based on the fine earth proportion declined in
both hypothetical (Fig. 2a) and field scenarios (Fig. 2b). This may be attributed to the fact that
the fine earth portion will get reduced proportionately with the gravel volume (Nagraja and
Srinivasamurthy, 2009; Grewal et al., 1984). Thus, both the hypothetical and the field
observations on soil organic C stocks remained the same in the order of Scenario 2 > Scenario 1
> Scenario 3.
**3.2     Extent of soil organic C variation**
The magnitude of deviations of soil organic C values in alternate estimation methods
(BDSM and FEV) from GSM were computed for both hypothetical (Fig. 2a) and field observed
(Figure 2b) values. Per cent deviations of the field sample observed soil organic C stocks were
calculated separately for both BDSM (Scenario 2) and FEV (Scenario 3) estimations (Fig. 3).
The per cent deviation values (Fig. 3) revealed a considerable matching of the predicted and the





observed values. This indicated that the observed differences between the GSM and the other
methods in the field estimations can be correlated with those of the hypothetical projections.
Comparative analysis of three different methods revealed that the soil organic C derived
using BD SM (Scenario 2) was found to be higher than the GSM (Scenario -1). The
accumulation of gravel in the furrow soil volume could add to the soil mass and BD values.
Contrastingly, the soil organic C values were low in the FEV method compared to the other 2
Scenarios. The differences among three estimates were found low in soils with lower gravel
levels and they increased with increase in the gravel content. Regression lines developed for
three estimates indicated that the differences increased with increase in gravel content.
Interestingly, the soil organic C estimates based on the FEV recorded higher $R^2$ values while the
BDSM estimations recorded the least. These observations suggested that the mass of the 'soil' in
the furrow layer is most critical in fertility estimations.
**3.3 Effect on available nutrients**
Similar observations were recorded for available nutrients (N, $P_2O_5$ and $K_2O$) (Fig. 4).
The available N (Fig. 4a), $P_2O_5$ (Fig. 4b), and $K_2O$ (Fig. 4c) revealed the same trends. The
explanation for these available nutrient relationships to method of estimation (GSM, BDSM and
FEV) could be traced to soil organic C trends. The available N, $P_2O_5$ and $K_2O$ stocks derived
using BDSM (Scenario 2) were found to be higher than the GSM method (Scenario 1). The
effects were due to the accumulation of gravel in the furrow soil volume, and adding to soil mass
and BD values. The available N, $P_2O_5$ and $K_2O$ values were least in FEV compared to other two
methods. The differences, as in soil organic C, were found to increase with increase in gravel
content. Increase in significance (GSM<BDSM<FEV) for these comparisons indicated the





impacts it might have on our evaluations based on the method of determination. As the stones do
not allow roots to grow and do not possess nutrient retention abilities, the nutrient estimations
based on the GSM or BDSM would only lead to over estimations of nutrients.
**4    Conclusions**
Our study indicates that the conventional methodology of using generalized soil mass
(GSM) or bulk density based soil mass (BDSM) in degraded soils would result in overestimation
of soil nutrients. These observations indicate that consideration of fine earth volume in the bulk
soil could be an important step in nutrient estimation, especially for physically degraded soils
with high gravel content. Selective accounting of fine earth portion is more applicable to both
moderately (15-35% gravel v/v) and gravely strong (35-85% gravel v/v) soils. The generalized
soil mass (GSM) based estimations could be well applicable for soils low in gravel. The extent of
variations in the three methods of estimation was low when gravel content is low. However, the
magnitude of variations among three estimation methods increases with increase in gravel
content as in degraded soils. Thus, selective accounting of fine earth portion in the bulk can be
adopted for realistic fertility estimations in degraded soils.
*Acknowledgements.* The authors thank Mr. Y.B. Srinivasa, Scientist, Institute of Wood Science
and Technology (IWST), Bangalore, and Dr. A Natarajan, National Bureau of Soil Survey and
Land Use Planning (NBSS&LUP), Bangalore, for providing valuable inputs and giving critical
comments in compiling this paper. Authors also thank Department of Soil Science, UAS,
Bangalore, for providing laboratory facility and ICAR, New Delhi for funding this research



project. We are highly thankful to the topical editor, Solid Earth, Dr. Artemi Cerda for useful suggestions for improving the manuscript further.

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

9    **Figure 3.** Deviation of the observed SOC values from the estimated values in different methods

10       of soil fertility estimation, where a) Based on BDSM (Bulk density based soil mass,

11       Scenario 2), and b) Based on FEV (Fine earth volume, Scenario 3).

12   **Figure 4.** Soil available N, $P_2O_5$, and $K_2O$ in relation to gravel content with GSM (Generalized

13       soil mass), BDSM (Bulk density based soil mass), and FEV (Fine Earth Volume) based.



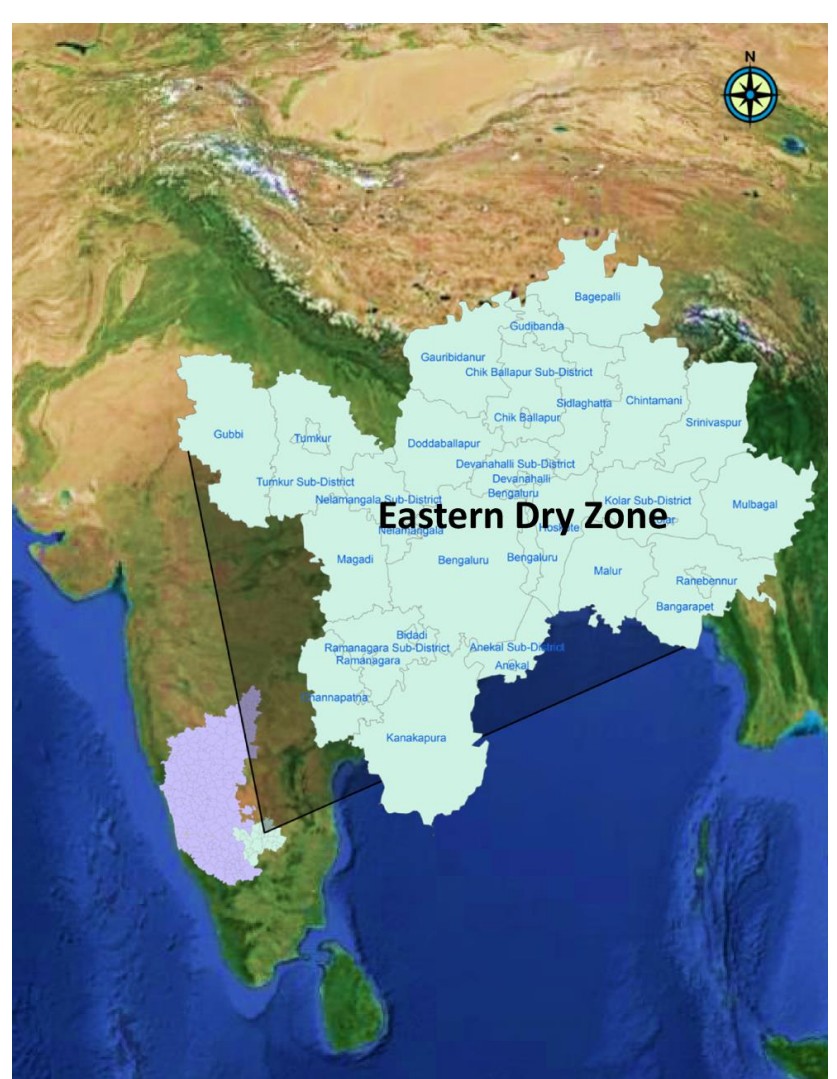

4      Figure 1







2     Figure 2





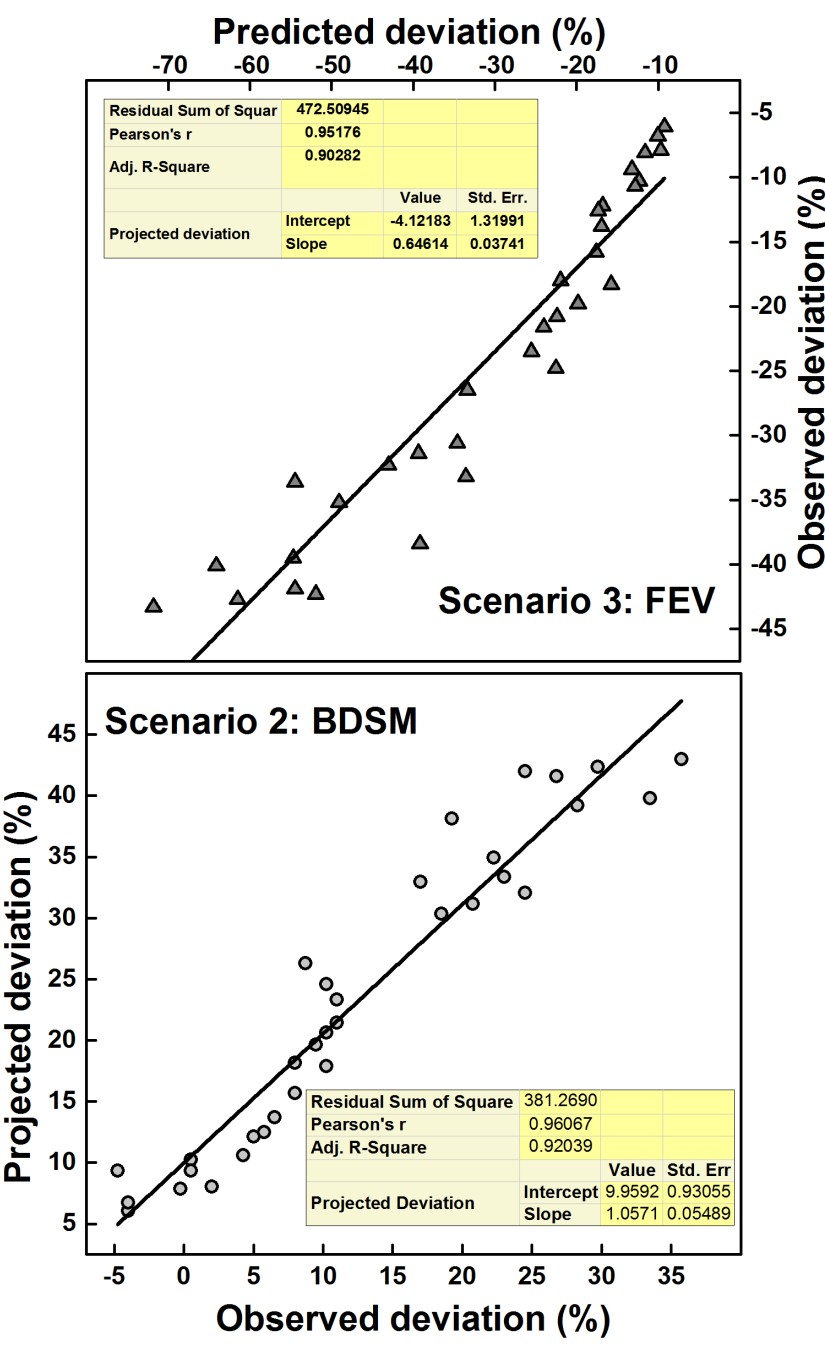

2      Figure 3





2    Figure 4