# Peer review of "Estimations of soil fertility in physically degraded soils through selective accounting of fine earth"

_Solid Earth, 2016_

## Referee Comment (RC1) · Anonymous Referee #1 · 9 Mar 2016

FIGURES: it could be possible to add colors to figures 2, 3 and 4? In the on-line version they are for free.

INTRODUCTION: These two references (Lozano-García et al., 2016 [Science of the Total Environment 544, 963-970. http://dx.doi.org/10.1016/j.scitotenv.2015.12.022] and Parras-Alcántara et al., 2015 [Land Degradation and Development 26, 800-806. http://dx.doi.org/10.1002/ldr.2231]) are more appropriate here (to support the idea of "help land developers and farmers to select management plans...") than Parras-Alcántara et al., 2015. Because the first one is related to natural areas with different kinds of vegetation [native or reforestated] and the second one is referred to agricultural lands under different managements.

MATERIAL AND METHODS: Please, clarify some aspects related to the sampling process and the methodology: 1. How many samples were taken in each sampling point? 2. How many replicas? 3. Could you locate the sampling points in the Figure 1? 4. Could you add a table with the main soil properties of the 18 soils sampled? 5. Between 2.2 and 2.3 sections I suggest including a new one in which you define the three methods (equation included) for the assessment of soil C and nutrient stocks.

RESULTS AND DISCUSSION: Please, be uniform respect to the use of acronyms. On the one hand, you use SOC in lines 12 and 13 in page 6 but you do not have defined this acronym previously, in fact you use soil organic C in the rest of the text. On the other hand, in the Abstract you refer to P and K, but in the 3.3 section you write P2O5 and K2O. Furthermore, you must improve and enrich the discussion by adding more current references.

---

## Referee Comment (RC2) · Anonymous Referee #2 · 12 Mar 2016

This paper showed a great efforts by the authors to measure soil organic C, available nitrogen (N), phosphorus (P) and potassium (K), and estimated stocks using three methods (generalized soil mass, bulk density based soil mass and the proportion of the fine earth volume) for soils from physically degraded lands in Eastern Dry Zone of Karnataka State in India. In this paper a relevant topic of soil science is worked in a non current study area. However, the methods did not present novel concepts or tools about soil properties in land degraded areas. I find the applied methods are correct and the obtained results are useful about a non typical study area (India). The results are sufficient to support the interpretation and conclusions, but the authors should work more in the discussion and to clear the applied methods. The description of the methods (soil collected samples and soil analysis are not clear for me). In conclusion, I think that the paper could be really interesting for the journal, but after some important revisions. Thank you very much for the

Firstly, I suggest general comments and finally, attached, the authors can observe some appreciations to improve and to reach, in my opinion, a higher scientific quality of the paper.

1) Title: I find the title is very clear, but maybe not very "interesting" or "attractive", because I think that in any moment the authors speak about soil fertility. The paper talks about an interesting correlation between the gravels and soil nutrients.

2) Key words: I didn't find this part.

3) Abstract: There aren't any explanations about where is developed the work and the aims.

4) Introduction: the authors must include more actual bibliography. Furthermore, there are more affirmations without citations. Finally, the aims of this work aren't clear, please, make a concrete paragraph only with the goals: i)...; ii)...; iii)...

5) Methods: Please, attach more information about the study area, soil samples (where, how many, why...). When you classify the soils, you must use actual and international "soil classifications", which all authors around the world can understand: USDA (2010) or FAO-WRB (2014). You should improve the statistical method description, because there are a lot of lakes about type of correlations, statistical programs...

6) Results: Please, make separated the results and discussion. In the results, you must improve the description of the results (correlations, numerical descriptions)... descriptions please, non conclusions or discussions. I recommend that you should use always the same nomenclature in all the text. In this moment, all is a little confuse: estimations/estimates; C/GSM; BD SM or BD???

7) Discussion: Please, put more attention in the author guidelines with the information about what is it a discussion. You should make a comparison between your results and others of different authors, and discuss methods, results and ideas. You need more bibliography. I suggest some references for the introduction too.

8) Figures: Figure number 1 needs coordinates, more information about soil sample locations. Figure 2, 3 and 4: I suggest to separate the correlations in tables, to clear the nomenclatures and the legend (symbols, lines. . .) and to reduce the decimals in the numbers.

[Figure]

**Supplement:**

[revised manuscript text omitted]

Figure 1

[Figure]

[Figure]

Figure 2

[Figure]

[Figure]

[Figure]

Figure 3

[Figure]

[Figure]

Figure 4

---

## Author Comment (AC1) · 9 May 2016

Anonymous Referee #1 Comments (Published: 9 March 2016)

COMMENT: FIGURES: it could be possible to add colors to figures 2, 3 and 4? In the on-line version they are for free.

REPLY: Agreed; the figures were changed to colour.

COMMENT: INTRODUCTION: These two references (Lozano-García et al., 2016 [Science of the Total Environment 544, 963-970. http://dx.doi.org/10.1016/j.scitotenv.2015.12.022] and Parras-Alcántara et al., 2015 [Land Degradation and Development 26, 800-806. http://dx.doi.org/10.1002/ldr.2231])

[Figure]

are more appropriate here (to support the idea of "help land developers and farmers to select management plans. . .") than ParrasAlcántara et al., 2015. Because the first one is related to natural areas with different kinds of vegetation [native or reforestated] and the second one is referred to agricultural lands under different managements.

REPLY: Agreed; the references were added to relevant sections.

COMMENT: MATERIAL AND METHODS: Please, clarify some aspects related to the sampling process and the methodology: 1. How many samples were taken in each sampling point? 2. How many replicas? 3. Could you locate the sampling points in the Figure 1? 4. Could you add a table with the main soil properties of the 18 soils sampled? 5. Between 2.2 and 2.3 sections I suggest including a new one in which you define the three methods (equation included) for the assessment of soil C and nutrient stocks.

REPLY: Agreed; new sections (section 2.3-stock estimation methods) were added in the manuscript describing samples taken, and methodology part was improved. The sampling points cannot be located on the map because of unavailability of GPS with student at the time of field work but a new table (Table 1) was added describing the general properties of degraded soils compared to normal soils in the area at eighteen sampled locations. A new section was added between 2.2 and 2.3, describing the three methods for the assessment of soil C and nutrient stocks, briefly. Other section numbers were accordingly changed.

COMMENT: RESULTS AND DISCUSSION: Please, be uniform respect to the use of acronyms. On the one hand, you use SOC in lines 12 and 13 in page 6 but you do not have defined this acronym previously, in fact you use soil organic C in the rest of the text. On the other hand, in the Abstract you refer to P and K, but in the 3.3 section you write $P_2O_5$ and $K_2O$. Furthermore, you must improve and enrich the discussion by adding more current references.

REPLY: Agreed; the acronyms were changes to maintain uniformity. SOC at L7 Page

6 and M&M Page 7 was changed to 'soil organic C' as used in all other sections of manuscript. P and K notations, in abstract and main body of the manuscript, were changed to $P_2O_5$ and $K_2O$, as represented in the graphs. Discussion was enriched with more relevant references, as suggested.

---

## Author Comment (AC2) · 9 May 2016

Anonymous Referee #2 Comments (Published: 12 March 2016)

Comment: This paper showed a great efforts by the authors to measure soil organic C, available nitrogen (N), phosphorus (P) and potassium (K), and estimated stocks using three methods (generalized soil mass, bulk density based soil mass and the proportion of the fine earth volume) for soils from physically degraded lands in Eastern Dry Zone of Karnataka State in India. In this paper a relevant topic of soil science is worked in a non current study area. However, the methods did not present novel concepts or tools about soil properties in land degraded areas. I find the applied methods are correct and the obtained results are useful about a non typical study area (India). The results

are sufficient to support the interpretation and conclusions, but the authors should work more in the discussion and to clear the applied methods. The description of the methods (soil collected samples and soil analysis are not clear for me). In conclusion, I think that the paper could be really interesting for the journal, but after some important revisions.

REPLY: Agreed; the methodology section has been further improved with addition of more information about sampling. New sections (section 2.3-Stock estimation methods) have been added in the methodology to provide further clarity. The discussion (as well as introduction) has been enriched with more references.

Comment: Firstly, I suggest general comments and finally, attached, the authors can observe some appreciations to improve and to reach, in my opinion, a higher scientific quality of the paper. 1) Title: I find the title is very clear, but maybe not very "interesting" or "attractive", because I think that in any moment the authors speak about soil fertility. The paper talks about an interesting correlation between the gravels and soil nutrients.

REPLY: The title has been slightly modified with addition of ". . .agricultural. . .and gravel fractions" which provides more insight into the contents of the paper. We hope this will be attractive enough to grab attention of reader interested in management of marginal lands with high gravel contenets.

Comment: 2) Key words: I didn't find this part.

REPLY: Done. "Key words" section has been added.

Comment: 3) Abstract: There aren't any explanations about where is developed the work and the aims.

REPLY: In abstract, line 9-10 explains where work was done, and lines 4-5 explain the aim of the study. We think this should be enough for abstract.

Comment: 4) Introduction: the authors must include more actual bibliography. Furthermore, there are more affirmations without citations. Finally, the aims of this work aren't

clear, please, make a concrete paragraph only with the goals: i). . .; ii). . .; iii). . .

REPLY: We agree that more references are appropriate and have added more references in introduction. This study was part of a larger set of data for management of marginal land areas in Eastern Dry Zone of Karnataka, India. The aim is clearly mentioned in Page 5 line 2-6 in the concluding paragraph of the introduction. A little re-wording was done to make it clear. We think it should serve the purpose.

Comment: 5) Methods: Please, attach more information about the study area, soil samples (where, how many, why. . .). When you classify the soils, you must use actual and international "soil classifications", which all authors around the world can understand: USDA (2010) or FAO-WRB (2014). You should improve the statistical method description, because there are a lot of lakes about type of correlations, statistical programs. . .

REPLY: Agreed; we have added a new section 2.3 (Stock estimation methods) in materials and methods and further clarified sampling methodology used (Page 6, Line 7-9). USDA, FAO/UNESCO classifications have also been added at page 5 line 14-15 for audience around the world. Statistical methods description has been further improved.

Comment: 6) Results: Please, make separated the results and discussion. In the results, you must improve the description of the results (correlations, numerical descriptions). . . descriptions please, non conclusions or discussions. I recommend that you should use always the same nomenclature in all the text. In this moment, all is a little confuse: estimations/estimates; C/GSM; BD SM or BD???

REPLY: We tried to make it easier by deleting some of the acronym which might cause confusion, like BD was for "bulk density" (soil characteristic) and BDSM was for "Bulk Density Based Soil Mass" (a method). BD acronym was deleted throughout text. For separation of results and discussion, we think this type of format best suited the work presented here. There are several papers in 'Solid Earth' with similar format yet conveying excellent messages. We await further directions from topical editor in this re-

gard, if change is required.

Comment: 7) Discussion: Please, put more attention in the author guidelines with the information about what is it a discussion. You should make a comparison between your results and others of different authors, and discuss methods, results and ideas. You need more bibliography. I suggest some references for the introduction too.

REPLY: We have added several new references to discussion section and tried to improve the discussion.

Comment: 8) Figures: Figure number 1 needs coordinates, more information about soil sample locations. Figure 2, 3 and 4: I suggest to separate the correlations in tables, to clear the nomenclatures and the legend (symbols, lines. . .) and to reduce the decimals in the numbers.

REPLY: Exact sample locations are not available since the student carrying survey had no availability of GPS but a new table with properties of marginal areas of all the sites compared to normal agricultural areas have been added.

Comment: References: Degraded areas by soil erosion, soils with high stoniness, influence of the vegetation and land cover, some predicting models. . . Cañadas, E.M., Jiménez, M.N., Valle, F., Fernández-Ondoño, E., Martín-Peinado, F., Navarro, F.B., 2010. Soil–vegetation relationships in semi-arid Mediterranean old fields (SE Spain): Implications for management. J. Arid Environ. 74, 1525–1533. doi:10.1016/j.jaridenv.2010.06.007 De Baets, S., Poesen, J., Meersmans, J., Serlet, L., 2011. Cover crops and their erosion-reducing effects during concentrated flow erosion. Catena 85, 237–244. doi:10.1016/j.catena.2011.01.009 Debolini, M., Schoorl, J.M., Temme, A., Galli, M., Bonari, E., 2013. CHANGES IN AGRICULTURAL LAND USE AFFECTING FUTURE SOIL REDISTRIBUTION PATTERNS: A CASE STUDY IN SOUTHERN TUSCANY (ITALY). Land Degrad. Dev. doi:10.1002/ldr.2217 Gabarrón-Galeote, M.A., Ruiz-Sinoga, J.D., Quesada, M.A., 2013. Influence of aspect in soil and vegetation water dynamics in dry Mediterranean conditions: functional adjustment of evergreen and semi-deciduous growth forms. Ecohydrology 6, 241–255. doi:10.1002/eco.1262 Hueso-González, P., Martínez-Murillo, J.F., Ruiz-Sinoga, J.D., 2014. The Impact of Organic Amendments on Forest Soil Properties Under Mediterranean Climatic Conditions. Land Degrad. Dev. 25, 604–612. doi:10.1002/ldr.2296 Imeson, A.C., Lavee, H., 1998. Soil erosion and climate change: the transect approach and the influence of scale. Geomorphology 23, 219–227. doi:10.1016/S0169-555X(98)00005-1 Likar, M., Vogel-Mikus, K., Potisek, M., Hancevic, K., Radic, T., Necemer, M., Regvar, M., 2015. Importance of soil and vineyard management in the determination of grapevine mineral composition. Sci. Total Environ. 505, 724–731. doi:10.1016/j.scitotenv.2014.10.057 Prosdocimi, M., Cerdà, A., Tarolli, P., 2016. Soil water erosion on Mediterranean vineyards: A review. Catena 141, 1–21. doi:10.1016/j.catena.2016.02.010 Qadir, M., Noble, A.D., Chartres, C., 2013. ADAPTING TO CLIMATE CHANGE BY IMPROVING WATER PRODUCTIVITY OF SOILS IN DRY AREAS. Land Degrad. Dev. 24, 12–21. doi:10.1002/ldr.1091 Rodrigo Comino, J., Brings, C., Lassu, T., Iserloh, T., Senciales, J., Martínez Murillo, J., Ruiz Sinoga, J., Seeger, M., Ries, J., 2015. Rainfall and human activity impacts on soil losses and rill erosion in vineyards (Ruwer Valley, Germany). SE 6, 823–837. doi:10.5194/se-6-823-2015 Ruiz-Sinoga, J.D., Diaz, A.R., 2010. Soil degradation factors along a Mediterranean pluviometric gradient in Southern Spain. Geomorphology 118, 359–368. doi:10.1016/j.geomorph.2010.02.003 Ruiz Sinoga, J.D., Martinez Murillo, J.F., 2009. Effects of soil surface components on soil hydrological behaviour in a dry Mediterranean environment (Southern Spain). Geomorphology 108, 234–245. doi:10.1016/j.geomorph.2009.01.012

REPLY: Suitable references have been added in the different sections in Introduction and Discussion.

Comment: Please also note the supplement to this comment: http://www.solid-earth-discuss.net/se-2016-26/se-2016-26-RC2-supplement.pdf

REPLY: The edits suggested/ issues raised in the supplement have been addressed

appropriately in the revised manuscript.